# A Combination of UV and Disinfectant for Inactivating Viable but Nonculturable State *Pseudomonas aeruginosa*: Efficiency and Mechanisms

**Jinfeng Zhao** [1,†], **Huichao Zhu** [1,†], **Chen Tao** [1], **Zhiquan Wang** [2], **Ning Deng** [1,*] and **Xin Huang** [1,*]

1. School of Environmental and Chemical Engineering, Shanghai University, Shanghai 200444, China; zjf10280@shu.edu.cn (J.Z.); zhuhuichao@shu.edu.cn (H.Z.); tc15255359512@163.com (C.T.)
2. School of Life and Environmental Science, Wenzhou University, Wenzhou 325035, China; 20210297@wzu.edu.cn
* Correspondence: dengning@shu.edu.cn (N.D.); huangxin2008@shu.edu.cn (X.H.); Tel.:+86-21-66137761 (N.D.); +86-21-66137748 (X.H.)
† These authors contributed equally to this work.

**Abstract:** Conventional disinfection techniques, relying on a single disinfection step, often fail to directly eliminate microorganisms, instead causing them to enter a viable but nonculturable (VBNC) state. However, microorganisms in the VBNC state retain metabolic activity and can reactivate under suitable conditions, representing a "hidden source of contamination" that threatens drinking water safety. This study fundamentally assessed the feasibility of combined disinfection methods by integrating $UV_{254}$ with disinfectant (NaClO, PAA, and PDS) for inactivating *Pseudomonas aeruginosa* (*P. aeruginosa*), an opportunistic pathogen that has been widely detected in water supply systems. The number of culturable cells was determined using the heterotrophic plate counting (HPC) method, and the number of VBNC cells was quantified using our recently developed qPCR approach. Quantitative analyses showed that combined disinfection methods can effectively reduce both culturable and VBNC cells by several orders of magnitude compared to a single disinfection step. Notably, VBNC *P. aeruginosa*, after 30 min of UV/NaClO treatment, was below the detection limit (3.191 log CFU/mL) of PMA-qPCR. The reactivation experiment also confirmed that VBNC *P. aeruginosa* did not reactivate for 16 h after 30 min of UV/NaClO treatment under controlled laboratory conditions. The higher disinfection capacity of combined methods can be attributed to the generation of reactive radicals. This study highlighted combined disinfection as a promising approach for the inactivation of bacteria in the VBNC state, yet further studies are needed before an application can be considered for minimizing VBNC reactivation in city utility water processing or high-risk building water distribution systems.

**Keywords:** *Pseudomonas aeruginosa*; free radicals; disinfection; viable but nonculturable; UV irradiation

## 1. Introduction

Bacterial contamination in drinking water has consistently remained a critical concern for public health. Pathogens such as *Escherichia coli* [1,2], *Salmonella* [3,4], *Vibrio cholerae* [5], and *Legionella pneumophila* [6] have attracted widespread attention, prompting the development of monitoring and prevention strategies. However, in recent years, threats to drinking water safety have expanded beyond traditional pathogens to include a broader range of environmental microbes, with *Pseudomonas aeruginosa* (*P. aeruginosa*) being a typical example [6–8]. *P. aeruginosa*, a common opportunistic pathogen capable of causing waterborne diseases [9–13], is prevalent in water supply systems [14], hospitals [15], pools [16], and other human activity sites. *P. aeruginosa* is prevalent in different settings, and significant proliferation has been observed in intensive care units (ICUs) and immunocompromised populations [17]. The inherent resistance mechanism of bacteria increases the infection rate,

and the resistance genes are acquired via horizontal gene transfer [18]. This increased resistance complicates the treatment of *P. aeruginosa* infections, thereby increasing healthcare costs and patient mortality.

Traditional disinfection methods are effective in controlling microbial populations in drinking water systems; however, the reactivation of *P. aeruginosa* after disinfection has been observed randomly. This phenomenon is attributed to the disinfection-induced viable but nonculturable (VBNC) state [19], wherein *P. aeruginosa* enters a dormant phase as a survival strategy against disinfection. In this state, bacteria exhibit reduced metabolic activity and evade detection by conventional culturing methods. However, this is not the end of the pathogen's lifecycle since VBNC *P. aeruginosa* can reactivate when environmental conditions become suitable [20], which becomes a concealed source of contamination in the drinking water supply.

Understanding the limitations of existing disinfection methods can realize an improvement in inactivating bacteria. Conventional drinking water treatment commonly employs a single disinfection method, such as ultraviolet ($UV_{254}$) [21], chlorination [22], persulfate (PDS) [23], and peracetic acid (PAA) [24]. Chlorine-based disinfectants exert oxidative stress on bacterial cells by disrupting their cell walls and membranes. This stress response triggers bacteria to enter the VBNC state, reducing metabolic activity and preserving cellular integrity [25]. $UV_{254}$ causes thymine dimers to form in the microbial DNA, rendering it incapable of replication. However, $UV_{254}$ cannot maintain a lasting disinfection effect, as this damage can be recovered via photoreactivation and dark repair [26]. Given that the mechanisms vary for different disinfection methods, the combination of a single disinfection method may offer synergistic effects.

The combination of $UV_{254}$ and disinfectant can generate reactive species that exhibit higher oxidative capacity, which has the potential to effectively inactivate bacteria. Wen et al. [27] demonstrated that UV/PDS accelerated the degradation of spore cell membranes and cell walls via the participation of hydroxyl radical ($OH^\bullet$) and sulfate radical ($SO_4^{\bullet-}$). Wang et al. [19] observed that UV/NaClO generated active chlorine species ($Cl^\bullet$, $NaClO^{\bullet-}$, and $ClO^\bullet$), which effectively reduced the metabolic activity and prevented the dark reactivation of *P. aeruginosa*. The accumulated evidence suggests that the combined process is expected to enhance the efficacy of *P. aeruginosa* inactivation while reducing the likelihood of reactivation. The synergistic effects primarily stem from the photolysis of the oxidant by $UV_{254}$. However, it is still unclear whether these combined disinfection methods can effectively inactivate *P. aeruginosa* in the VBNC state.

Quantification of *P. aeruginosa* in the VBNC state presents another challenge for evaluating disinfection efficiency. Traditional methods rely on the culturable ability of bacteria to grow and form colonies on agar plates, which underestimates the presence of VBNC cells to grow under standard laboratory conditions. To address this challenge, we recently developed alternative techniques for quantifying VBNC *P. aeruginosa* via the integration of propidium monoazide (PMA) with a quantitative polymerase chain reaction (qPCR) approach [28]. PMA selectively penetrates cells with damaged membranes and binds to their DNA, preventing its amplification during qPCR. Hence, the combination of PMA-qPCR with heterotrophic plate counting (HPC) allowed us to quantify the total number of viable *P. aeruginosa*. Subtracting HPC-detected cells (i.e., culturable cells) from the viable cells identified via qPCR yields the VBNC population. This innovative method allows us to evaluate the efficiency of the combined disinfection methods for inactivating VBNC *P. aeruginosa*.

This study aims to evaluate the feasibility of the combined processes (i.e., UV/NaClO, UV/PDS, and UV/PAA) for inactivating *P. aeruginosa*. To achieve this goal, PMA-qPCR and HPC were employed to ascertain the overall count of viable and culturable cells of *P. aeruginosa*, and the difference between the two is the number of cells in the VBNC state. Probe and quenching experiments were employed to investigate the disinfection mechanisms of the combined processes. Reactivation experiments were also performed to assess the microbial stability after water disinfection. It is worth noting that this study

used longer gene segments in PMA-qPCR, which effectively mitigated false amplification caused by short DNA sequences, thereby improving the accuracy of quantifying viable cell counts. Moreover, we applied the method to evaluate various combined disinfection treatments, including UV/NaClO, UV/PDS, and UV/PAA treatments. Furthermore, we studied the influence of surface water turbidity on disinfection effectiveness, thus providing new insights into practical efforts in combined disinfection treatments.

## 2. Materials and Methods

### 2.1. Laboratory Setting

The study was conducted in an academic laboratory equipped with advanced ventilation systems and biological safety cabinets. Laboratory personnel include research and technical staff who have undergone professional training and hold relevant degrees. They receive regular training in biosafety and laboratory procedures, are familiar with safe handling procedures for pathogenic microorganisms, and strictly adhere to laboratory safety regulations. During experiments, personnel wear appropriate personal protective equipment and operate within biological safety cabinets. All consumables used in disinfection experiments are autoclaved at 121 °C to ensure no bacterial contamination. The air inside the laminar flow hood is filtered by a high-efficiency filter and sterilized by $UV_{254}$ before the experiment to ensure that there is no other microbial contamination. The parameters of the incubator are set to optimize the growth conditions for *P. aeruginosa* to ensure its viability. All laboratory equipment used for research (e.g., centrifuges and $UV_{254}$ lamps) is tested and calibrated to meet the requirements of the experiment before conducting research to collect data. After experiments, all samples are properly handled and disposed of to prevent any potential contamination or hazards.

### 2.2. Reagents

*P. aeruginosa* (Collection No. CMCC(B)10104) was obtained from HuanKai Biology Co., Ltd. (Guangzhou, China) and cultured aseptically in Luria–Bertani (LB) medium. All reagents were analytical grade unless otherwise noted. Potassium persulfate (PDS, 99.0%) was purchased from Aladdin BioChem Technology Co., Ltd., Shanghai, China. Sodium hypochlorite (NaClO), peracetic acid (PAA), metronidazole (MET), nitrobenzene (NB), tert-butanol (TBA), and ethanol (EtOH) were purchased from Sinopharm Chemical Reagent Co., Ltd. (Shanghai, China). All solutions were prepared with Milli-Q water ($\geq 18.2$ MΩ · cm). Luria broth (LB) and nutrient agar (NA) were purchased from Circle K Biotechnology Co., Ltd., Shanghai, China. Propidium monoazide (PMA) was obtained from Biotium USA, Inc. DNeasy®PowerSoil® Pro Kit was purchased from Qiagen USA, Inc. SYBR®Green PCR Mix was obtained from Yirui Biotechnology Co., Ltd., Shanghai, China. Primers, dd $H_2O$, and DNA Markers were purchased from Sangon Bioengineering Co., Shanghai, China. Phosphate buffer solution (PBS, pH = 7.0, Arlington, VA, USA) was prepared from NaCl, $Na_2HPO_4$ and $NaH_2PO_4$.

### 2.3. Cultivation of P. aeruginosa Cells

The bacterial strain was inoculated into an LB culture medium and grown at 37 °C, 150 rpm for 18 h. The bacteria were centrifuged in a high-speed refrigerated centrifuge (4 °C, 2320× *g*) for 10 min. Then, the supernatant was discarded, and the bacteria were washed with sterile PBS (pH = 7.0). The above steps were repeated twice, and the resulting bacteria were resuspended in PBS solution. The absorbance of washed cell suspension was measured at 600 nm to achieve a consistent initial cell density (~$3.32 \times 10^7$ CFU/mL) in the reactor.

### 2.4. Experimental Setup of the Combined Disinfection Processes

A UV reactor was utilized within a parallel beam instrument. $UV_{254}$ was generated using a commercially available mercury lamp (Heraeus, Hanau, Germany) with a wavelength of 254 nm. UV light refers to the emission of light from UV lamps, while UV irradiation

denotes the exposure of a substance to UV light. It is important to note that UV light encompasses a range of wavelengths, including 222 nm, 254 nm, 275 nm, and 405 nm. Previous studies have reported that UV at 254 nm can disrupt microbial DNA and has been widely used in real applications. In this study, $UV_{254}$ represents the UV treatment method [29]. The $UV_{254}$ dose selected in this paper is higher than the actual radiation dose used in the water treatment scenario (40 mJ/cm$^2$). The chosen dose was deliberately elevated to facilitate the inactivation of *P. aeruginosa* for research purposes. To ensure the stability of photon output, the UV lamp underwent a 30 min warm-up before the experiment. The bacterial suspension was added to a sterile Petri dish in 40 mL sterile water to achieve an initial concentration of *P. aeruginosa* at $3.32 \times 10^7$ CFU/mL. The Petri dish was then placed on a magnetic stirrer with the addition of 40 μM NaClO, 40 μM PDS, or 40 μM PAA. The turbidity of the surface water was found to range from 7.59 to 44.5 NTU in Shanghai, China (Figure S1). In this study, montmorillonite was used to simulate the particles in water, the water body is configured with 40 NTU. Samples were collected at different time intervals of 0, 5, 10, 15, 20, 25, and 30 min. After sampling, the reaction was immediately quenched by adding excess sodium thiosulfate (0.1 mM) for the subsequent analysis. Each disinfection experiment was conducted with three parallel replicates.

### 2.5. Quantification of the Culturable and VBNC Cells

The number of culturable bacteria was determined using the HPC method. Initially, 1.0 mL samples were collected and diluted with sterile water. Subsequently, the diluted samples were inoculated onto nutrient agar plates and incubated at 37 °C for 24 h. Finally, the colony counts for each plate were recorded, and the colony-forming units (CFU) per mL of the original sample were calculated based on the dilution factor and the number of colonies observed.

PMA-qPCR was employed to quantify the overall count of viable cells using the long primers (*opr*-L, Table S1). The validation of PMA-qPCR was elucidated in our previous publication [28]. During PMA-qPCR measurement, a stock solution of 2.5 mM PMA was first prepared and stored at −20 °C. A 10 μL PAA solution was added to a 1 mL sample containing *P. aeruginosa*, followed by thorough mixing and incubation in a centrifuge tube under dark conditions for 15 min. Then, the centrifuge tube was placed on ice and exposed to a 650 W halogen lamp (Changsheng Special Light Source Electric Appliance Factory) for 15 min. After exposure, the sample was filtered through a 0.22 μm membrane, and bacterial DNA was extracted for the qPCR analysis using the DNeasy® PowerSoil® Pro Kit reagents.

SYBR Green qPCR experiments were performed in three parallel groups on the Bio-Rad Cycler CFX96 (CFX96 Touch™ Deep Well Real-Time PCR Detection System, Bio-Rad, Munich, Germany). The reaction system consisted of 1 μL template DNA, 0.4 μL each primer (10 μM), 10 μL SYBR ® Green PCR Mix, and 8.2 μL H$_2$O. The qPCR program consisted of 1 cycle at 95 °C for 5 min for DNA polymerase activation, followed by 40 cycles consisting of 95 °C for 15 s and 60 °C for 45 s for primer annealing and elongation. A melting curve, ranging from 60 °C to 95 °C with a step of 0.5 °C/second, was automatically generated in the system. All measurements were performed in triplicates, and the average values and standard deviations were reported in this study.

The population of VBNC cells can be calculated by subtracting the number of culturable cells determined by HPC from the viable cells identified via PMA-qPCR.

### 2.6. Bacterial Reactivation Experiment

An amount of 10 mL VBNC state *P. aeruginosa* was placed into a centrifuge tube containing an equal volume of sterile LB broth. The sample was incubated at 37 °C with continuous shaking at 150 rpm. Every 2 h interval, the cultivability of the sample was identified using the HPC method until culturable bacteria became apparent, allowing for the determination of lag period and recovery quantity. After disinfection, reactivation experiments were conducted in ten parallel sets, with microbial counts performed in triplicate for each set of reactivation experiments.

### 2.7. Application of Quenchers with Radicals

Tert-butanol (TBA) and ethanol (EtOH) were used as quenching agents to evaluate the respective contributions of HO$^\bullet$ and SO$_4^{\bullet-}$ to bacterial inactivation. TBA has a rate constant of 3.8–7.6 × 10$^8$ M$^{-1}$ s$^{-1}$ with HO$^\bullet$, much higher than 4.0–9.1 × 10$^5$ M$^{-1}$ s$^{-1}$ with SO$_4^{\bullet-}$, which was utilized as a highly effective scavenger for HO$^\bullet$. EtOH was employed with a rate constant of 1.2–2.8 × 10$^9$ M$^{-1}$ s$^{-1}$ and 1.6–7.7 × 10$^7$ M$^{-1}$ s$^{-1}$ for scavenging SO$_4^{\bullet-}$ and HO$^\bullet$, which was utilized as a highly effective scavenger for SO$_4^{\bullet-}$.

Nitrobenzene (NB) was selected as an organic probe for HO$^\bullet$, as its reaction rate with HO$^\bullet$ ($k_{HO^\bullet, NB}$ = 3.9 × 10$^9$ M$^{-1}$ s$^{-1}$) is much higher than that with sulfate radicals ($k_{SO_4^{\bullet-}, NB}$ < 10$^6$ M$^{-1}$ s$^{-1}$, Table S2). Therefore, the steady-state concentration of HO$^\bullet$ ([HO$^\bullet$]$_{ss}$) in the reaction, systems could be determined by measuring the degradation of NB. In addition, metronidazole (MET) was used as an organic probe that can react with both HO$^\bullet$ and SO$_4^{\bullet-}$ at similar reaction rates. The reaction rate of MET with HO$^\bullet$ ($k_{HO^\bullet, MET}$ = 3.8 × 10$^9$ M$^{-1}$ s$^{-1}$) and sulfate radicals ($k_{SO_4^{\bullet-}, MET}$ = 8.4 × 10$^9$ M$^{-1}$s$^{-1}$, Table S2) allows for the calculation of the steady-state concentration of SO$_4^{\bullet-}$ ([SO$_4^{\bullet-}$]$_{ss}$) in the reaction systems, while [HO$^\bullet$]$_{ss}$ was already determined using NB degradation.

The degradation kinetics of NB can be calculated by using the following equation:

$$k_{HO^\bullet, \text{NB}} [HO^\bullet]_{ss} = k_{\text{obs, NB}} \tag{1}$$

Similarly, the steady-state concentration of SO$_4^{\bullet-}$ ([SO$_4^{\bullet-}$]$_{ss}$) can be determined by utilizing the degradation of MET, together with the previously determined [HO$^\bullet$]$_{ss}$.

$$k_{HO\bullet, \text{MET}} [HO^\bullet]_{ss} + k_{SO_4^{\bullet-}, \text{MET}} [SO_4^{\bullet-}]_{ss} = k_{\text{obs, MET}} \tag{2}$$

### 2.8. Estimation of the Contributions of UV/Disinfectant to Inactivate P. aeruginosa

The contribution rate of various mechanisms to the comprehensive inactivation of *P. aeruginosa* was estimated by the *P. aeruginosa* inactivation rate constant, expressed as $k_{UV}$ and $k_{(\text{NaClO/PDS/PAA})}$. The inactivity mechanics of all methods (UV$_{254}$, NaClO, PDS, PAA, etc.) follow the pseudo-first-order model, and the corresponding coefficient of determination ($R^2$) is shown in Figure S2. These constants were derived from fitting bacterial inactivation curves with Equations (3) and (4). In addition, $k_{\text{radicals}}$ were calculated via the decreased levels after quenching radicals by TBA, as shown in Equation (5).

$$-\log\left(\frac{N_t}{N_0}\right)_{\text{inactivation}}$$
$$= -\log\left(\frac{N_t}{N_0}\right)_{UV} - \log\left(\frac{N_t}{N_0}\right)_{(\text{NaClO/PDS/PAA})}$$
$$-\log\left(\frac{N_t}{N_0}\right)_{\text{radicals}} - \log\left(\frac{N_t}{N_0}\right)_{\text{synergistic effect}} \tag{3}$$

$$k_{\text{inactivation}} = k_{UV} + k_{\text{NaClO/PDS/PAA}} + k_{\text{radicals}} + k_{\text{synergistic effect}} \tag{4}$$

$$k_{\text{radicals}} = k_{\text{inactivation}} - k_{\text{inactivation+TBA}} \tag{5}$$

## 3. Results

### 3.1. Enhanced Inactivation of P. aeruginosa by Combination Disinfection

The effectiveness of inactivating *P. aeruginosa* by combined disinfection strategies (UV/NaClO, UV/PDS, and UV/PAA) was evaluated by quantifying the number of culturable and viable cells, in comparison with the control experiment by a single disinfection step (i.e., UV$_{254}$, NaClO, PDS, and PAA). The number of culturable cells was determined using HPPC, as shown in Figure 1a, and the number of viable cells was quantified based on the membrane integrity using the PMA-qPCR method (Figure 1b). For the single disinfection step, UV$_{254}$ (26.1 mJ/cm$^2$) and NaClO (40 μM) treatment achieved complete elimination of the curable cells in approximately 3 min and 4 min. PAA reduced the number of culturable cells by four orders of magnitude (4-log10), while PDS had negligible effects on the inactiva-

tion of culturable cells. Notably, the combined disinfection strategies significantly enhanced the inactivation of *P. aeruginosa*. For all conditions, the culturable bacteria decreased to 0 CFU/mL within 1 min.

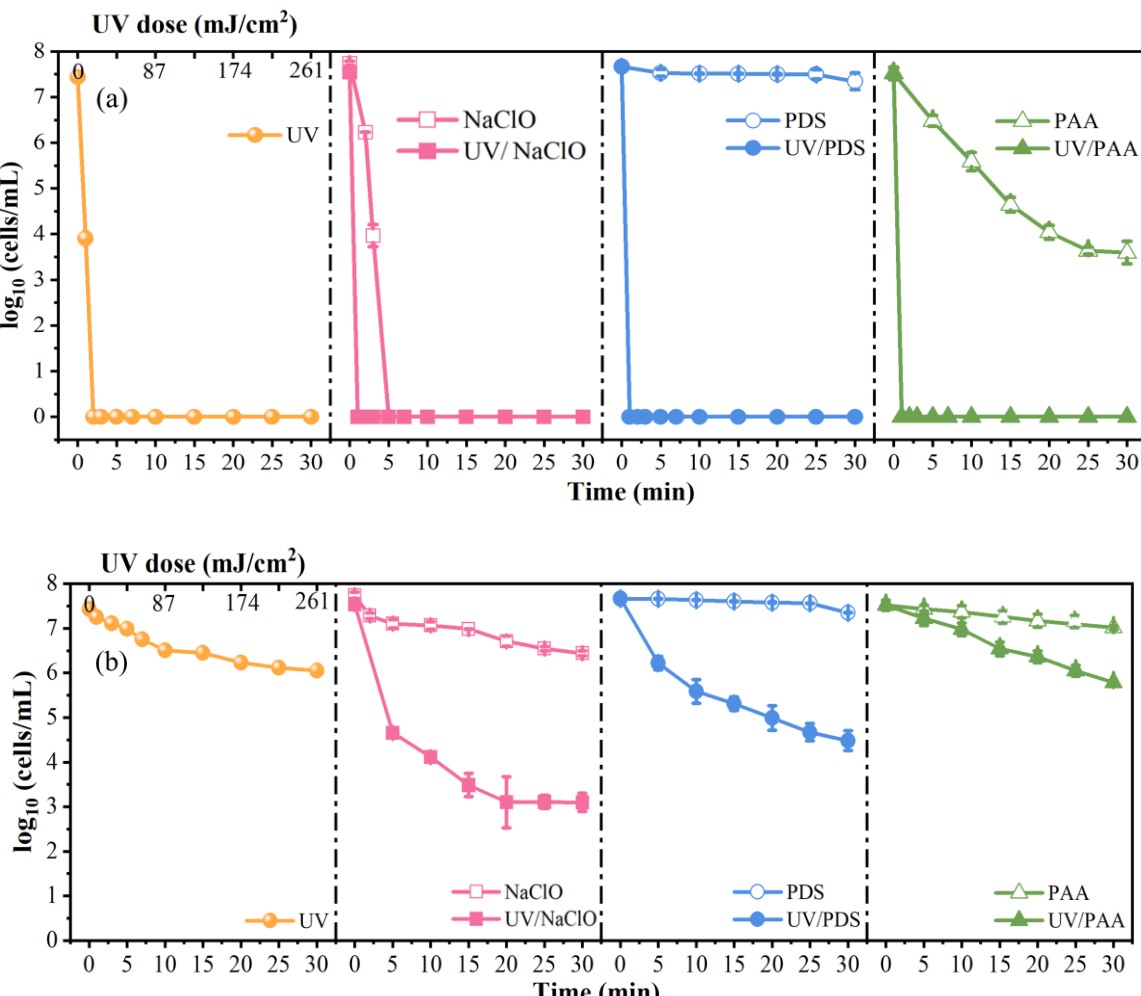

**Figure 1.** The number of culturable *P. aeruginosa* in PBS during UV, NaClO, PDS, PAA, UV/NaClO, UV/PDS, and UV/PAA determined by the HPC method (**a**), and the number of viable *P. aeruginosa* in PBS during UV, NaClO, PDS, PAA, UV/NaClO, UV/PDS, and UV/PAA determined by the PMA-qPCR method (**b**). Please note that the detection limit for viable cells using the PMA-qPCR method was 3.19 log CFU/mL. Conditions: $[NaClO]_0 = [PDS]_0 = [PAA]_0 = 40$ μM, UV fluence rate = 0.145 mW/cm$^2$, initial concentration of *P. aeruginosa* = $3.32 \times 10^7$ CFU/mL, pH = $7.0 \pm 0.2$, T = $25 \pm 2$ °C.

The scenario was not the same for the viable cells during disinfection treatments. For the single disinfection step, there was no obvious decrease in the number of viable cells, with a slight decrease of less than 1.5 log CFU/mL. Compared to a significant decrease in the culturable cells by $UV_{254}$ or NaClO treatment, this difference indicated that the majority of cells entered the VBNC state when exposed to the single disinfection step. However, the combined disinfection strategies notably decreased the viable cells. The number of VBNC *P. aeruginosa* after UV/NaClO treatment was below the detection limit (3.191 log CFU/mL) of PMA-qPCR, which indicated that UV/NaClO treatment can effectively control *P. aeruginosa* from entering the VBNC state compared to UV/PDS and UV/PAA treatment. For the treatment by UV/PDS and UV/PAA, viable cells decreased from 7.44 log CFU/mL to $4.48 \pm 0.22$ log CFU/mL and $5.79 \pm 0.03$ log CFU/mL, respectively. Since no culturable cells were observed in UV/PDS or UV/PAA treatments, the remaining viable cells after UV/PDS ($4.48 \pm 0.22$ log CFU/mL) or UV/PAA ($5.79 \pm 0.03$ log CFU/mL) could indicate

that a significant amount of *P. aeruginosa* entered the VBNC state. In summary, UV/NaClO exhibited a stronger capability to completely eliminate *P. aeruginosa* without the formation of VBNC cells, whereas UV/PDS and UV/PAA substantially reduced culturable cells but also posed the risk of inducing *P. aeruginosa* into a VBNC state.

### 3.2. Inactivation Mechanisms

The enhanced performance of combined disinfection for the inactivation of *P. aeruginosa* was supposed to be associated with the generation of reactive species during UV-activated advanced oxidation processes (AOPs). To test this possibility, free radical quenching experiments were conducted using tert-butanol (TBA, 50 mM) as a free radical scavenger [30]. Preliminary experiments showed that the sole addition of 50 mM TBA did not affect the activity of *P. aeruginosa* (Figure S4) [31,32]. In the presence of 50 mM TBA at pH 7 (Figure 2a), the inactivation of viable cells was suppressed by UV/NaClO, UV/PDS, and UV/PAA. The inactivation rate decreased by 58.18%, 58.83%, and 55.54% (Figure 2b), respectively, indicating the significant role of free radicals in the UV-AOPs inactivation of *P. aeruginosa*.

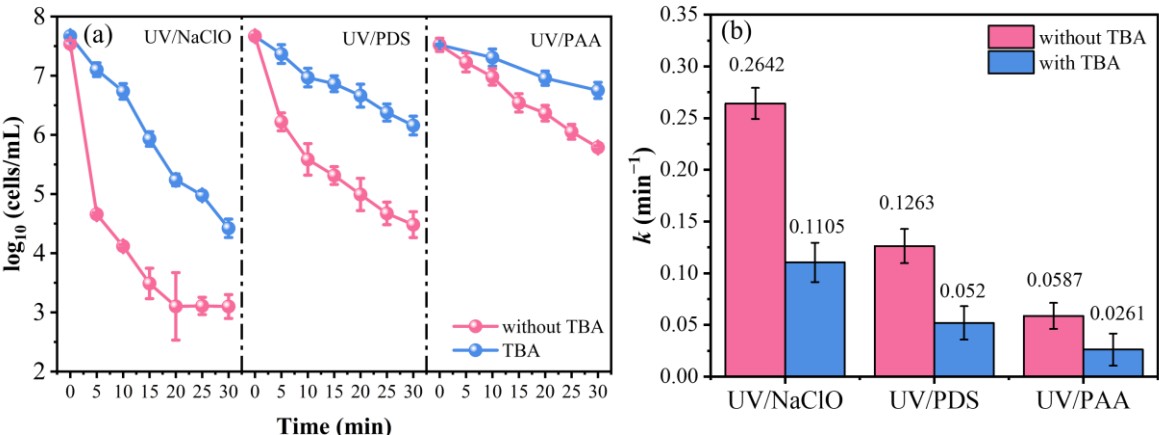

**Figure 2.** Inactivation curves of *P. aeruginosa* by the combined disinfection methods with and without TBA (**a**) and inactivation rate constants of *P. aeruginosa* (**b**). Conditions: $[NaClO]_0 = [PDS]_0 = [PAA]_0 = 40\ \mu M$, $[TBA]_0 = 50$ mM, UV fluence rate = 0.145 mW/cm$^2$, initial concentration of *P. aeruginosa* = $3.32 \times 10^7$ CFU/mL, pH = 7.0 ± 0.2, T = 25 ± 2 °C.

Nitrobenzene (NB) and metronidazole (MET) were selected as probe agents to quantify the concentrations of $[HO^\bullet]ss$ and $[SO_4^{\bullet-}]ss$ in UV-AOPs [33,34]. The concentration changes in NB and MET in the UV/NaClO, UV/PDS, and UV/PAA systems were measured using ultra-performance liquid chromatography (UPLC, Tables S3 and S4). Results showed that the decay of NB and MET followed pseudo-first-order kinetics over time (Figure S5). The free radical concentrations in each system were calculated according to the steady-state free radical calculation equations (Equations (1) and (2)), as shown in Table 1.

**Table 1.** Concentrations of HO• and SO$_4$•$^-$ in the UV/NaClO, UV/PDS, and UV/PAA processes at pH 7.

| | Quantum Yield | Free Radical Generation Efficiency (This Study) | |
|---|---|---|---|
| | | **[HO•]ss (M)** | **[SO$_4$•$^-$]ss (M)** |
| UV/NaClO | 1.0 mol Es$^{-1}$ [35] | $5.32 \times 10^{-13}$ | |
| UV/PDS | 0.7 mol Es$^{-1}$ [35] | $2.92 \times 10^{-13}$ | $9.04 \times 10^{-14}$ |
| UV/PAA | 0.88 mol Es$^{-1}$ [36] | $5.40 \times 10^{-13}$ | |

Conditions: $[NaClO]_0 = [PDS]_0 = [PAA]_0 = 40\ \mu M$, [NB] = [MET] = 0.5 μM, UV fluence rate = 0.145 mW/cm$^2$, initial concentration of *P. aeruginosa* = $3.32 \times 10^7$ CFU/mL, pH = 7.0 ± 0.2, T = 25 ± 2 °C.

The concentrations of [HO$^\bullet$]ss in UV/NaClO (5.32 $\times$ 10$^{-13}$ M) and UV/PAA (5.40 $\times$ 10$^{-13}$ M) were almost identical (Table 1). However, UV/NaClO treatment demonstrates notable efficacy in the inactivation of *P. aeruginosa*. This discrepancy can be elucidated from two perspectives: firstly, the UV/NaClO treatment exhibits the highest quantum yield and free radical generation efficiency; secondly, it facilitates the generation of active chlorine species (RCS), including Cl$^\bullet$, ClO$^\bullet$, and NaClO$^{\bullet-}$. Although the concentration of [HO$^\bullet$]ss in UV/PDS is lower relative to UV/PAA, the higher inactivation efficiency of *P. aeruginosa* in UV/PDS than that in UV/PAA might be mainly due to the generation of SO$_4^{\bullet-}$ by PDS under UV irradiation. The oxidation potential of SO$_4^{\bullet-}$ (2.5–3.1 V) exceeds that of HO$^\bullet$ (2.0–2.8 V), which is conducive to more effective oxidation and degradation of bacterial cell membranes.

It was also noteworthy that quenching radicals did not completely inhibit the inactivation of *P. aeruginosa* in UV-AOPs. Therefore, other inactivation mechanisms, including (1) direct exposure to UV$_{254}$; (2) catalysis by NaClO, PDS, and PAA leading to direct oxidation; and (3) synergies effects, may participate in the inactivation of *P. aeruginosa*. A strict differentiation of the contribution from each mechanism is extremely challenging. In this study, the contribution from each mechanism was roughly estimated by calculating the inactivation rate (*k* value) in a sequence of control experiments. In UV/NaClO, the contributions of different mechanisms to the inactivation of *P. aeruginosa* were summarized as free radicals (58%) > UV(21%) > NaClO (18%) > synergistic effect (3%). Compared to the contributions of PDS to the UV/PDS (3%) and PAA to the UV/PAA (16%) systems, NaClO plays a more substantial role in the overall inactivation efficiency of the UV/NaClO system (18%). This observation could be attributed to the capacity of NaClO to induce more severe damage to the cellular structure (Figure 3). In summary, the observed collaboration in UV-AOPs can be estimated via the engagement of multiple damage mechanisms. UV$_{254}$ primarily impacts DNA, whereas chemical oxidants like NaClO and PDS are presumed to impair microbial cell walls and microbial cell membranes, as well as enzymes or transport systems. Additionally, PAA can deactivate catalase, thereby amplifying the disinfection impact of free radicals on *P. aeruginosa*.

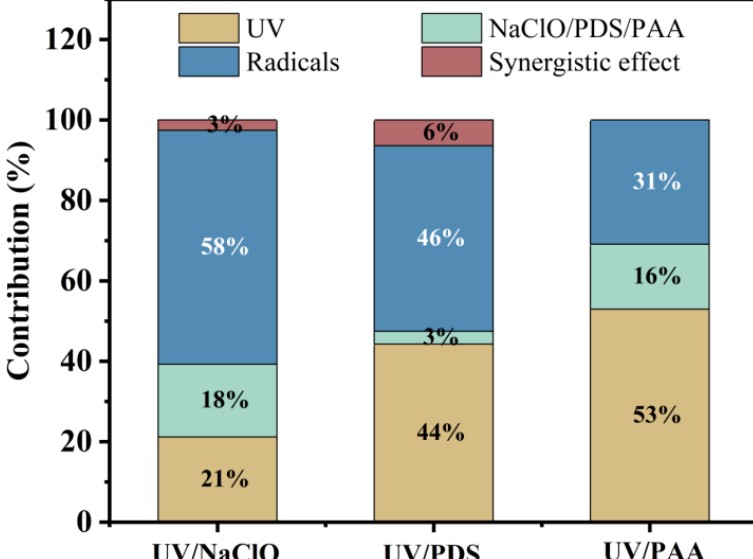

**Figure 3.** Contributions of UV, NaClO/PDS/PAA, radicals, and synergistic effect to inactivation of *P. aeruginosa* under UV/NaClO, UV/PDS, and UV/PAA. Conditions: [NaClO]$_0$ = [PDS]$_0$ = [PAA]$_0$ = 40 µM, UV fluence rate = 0.145 mW/cm$^2$, initial concentration of *P. aeruginosa* = 3.32 $\times$ 10$^7$ CFU/mL, pH = 7.0 $\pm$ 0.2, T = 25 $\pm$ 2 °C.

### 3.3. Assessing the Risk of Reactivation after UV/NaClO, UV/PDS, and UV/PAA

To evaluate the long-term inactivation effect and safety of UV-AOPs, the monitoring of reactivation potential was conducted for 16 h. The recovery of *P. aeruginosa* was monitored after 30 min of treatment with UV/NaClO, UV/PDS, and UV/PAA. *P. aeruginosa* counts were maintained at 0.0 log CFU/mL for up to 16 h under controlled laboratory conditions (Figure 4a). This level of detection suggests that combined UV/NaClO disinfection inactivated VBNC *P. aeruginosa* without VBNC cell formation for up to 16 h. However, in UV/PDS (Figure 4b) and UV/PAA (Figure 4c), reactivation of VBNC *P. aeruginosa* occurred at 10 h and 14 h, respectively, with the *P. aeruginosa* count rapidly rising to 1.95–2.08 log CFU/mL and 2.08–2.30 log CFU/mL. This indicated that part of *P. aeruginosa* entered a VBNC state with UV/PDS or UV/PAA treatment. The reactivation risk level after the combined treatment was UV/PDS > UV/PAA > UV/NaClO, which was because *P. aeruginosa* entered the VBNC state under the UV/PDS and UV/PAA treatments. It is worth noting that the UV/NaClO treatment has the highest disinfection effect and therefore the lowest risk of reactivation. Unlike PDS, which primarily damages the external structure of the cell, PAA works by penetrating the cell and producing active free radicals in the cell [37], which may lead to a lower risk of reactivation. This observation suggests a correlation between bacterial reactivation and cell integrity that warrants further investigation [19].

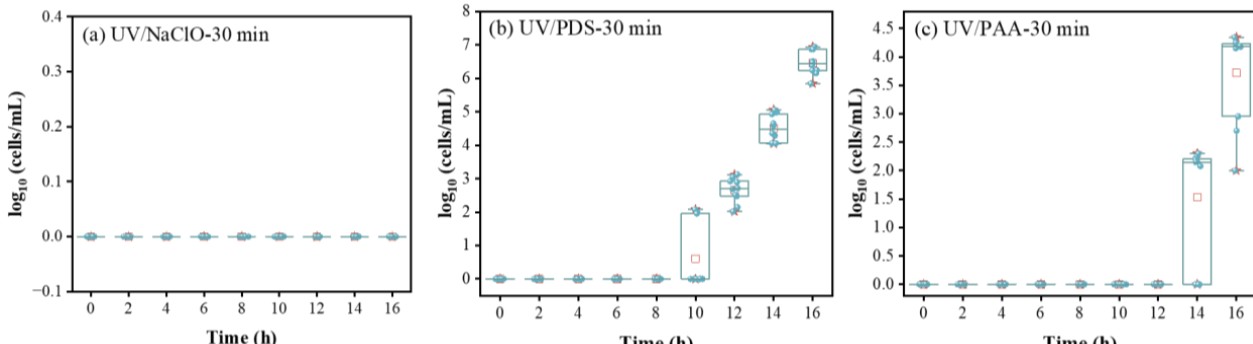

**Figure 4.** Reactivation (time in hours) of *P. aeruginosa* after 30 min of treatment with (**a**) UV/NaClO, (**b**) UV/PDS, and (**c**) UV/PAA. Conditions: $[NaClO]_0 = [PDS]_0 = [PAA]_0 = 40$ μM, UV fluence rate = 0.145 mW/cm$^2$, initial concentration of *P. aeruginosa* = $3.32 \times 10^7$ CFU/mL, pH = $7.0 \pm 0.2$, T = $25 \pm 2$ °C.

### 3.4. Assessing the Effects of Water Matrices on Inactivation

In drinking water treatment, the turbidity may affect the disinfection performance of *P. aeruginosa*. To explore this possibility, we configured water with a turbidity of 40 NTU to investigate the inactivation rate of *P. aeruginosa* (Figure 5a). In UV/NaClO, the increase in turbidity from 0 NTU to 40 NTU led to the inactivation rate dropping from 0.2642 min$^{-1}$ to 0.1495 min$^{-1}$. In UV/PDS, the inactivation rate decreased from 0.1263 min$^{-1}$ to 0.0791 min$^{-1}$. In UV/PAA, the inactivation rate decreased from 0.0587 min$^{-1}$ to 0.0396 min$^{-1}$ (Figure 5b). The decrease in the inactivation rate was mainly due to the decrease in $UV_{254}$ efficiency caused by the increase in water turbidity, which led to a decrease in the inactivation rate of *P. aeruginosa* [38]. As the water turbidity increased, the scattering and absorption of light increased, diminishing the depth to which $UV_{254}$ could penetrate and consequently reducing the efficiency of $UV_{254}$ [39]. In addition, the increase in water turbidity would also reduce the exposure area of the bacterial surface to $UV_{254}$, thereby further reducing the efficiency of UV radiation. Although the increase in water turbidity reduced the inactivation efficiency of UV/NaClO, the number of VBNC cells was reduced to reach the detection limit (3.191 log CFU/mL) upon increasing the UV dose. This suggested that extending the increasing UV dose can compensate for the negative effects of the turbidity on inactivating *P. aeruginosa*.

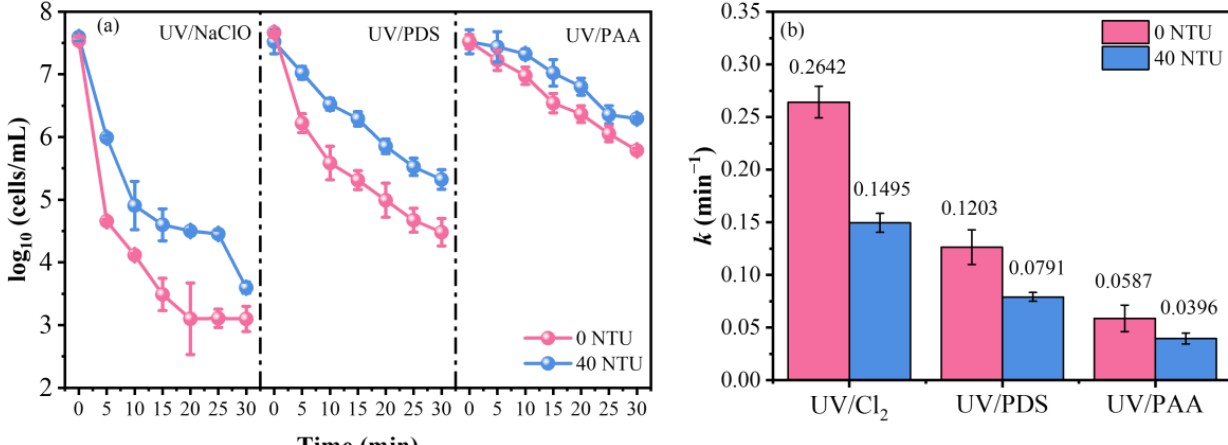

**Figure 5.** Effect of turbidity (40 NTU) on the inactivation curve (**a**) and inactivation rate constant (**b**) of *P. aeruginosa* by the treatment with UV/NaClO, UV/PDS, or UV/PAA. Conditions: $[NaClO]_0 = [PDS]_0 = [PAA]_0 = 40$ μM, UV fluence rate = 0.145 mW/cm$^2$, initial concentration of *P. aeruginosa* = $3.32 \times 10^7$ CFU/mL, pH = 7.0 ± 0.2, T = 25 ± 2 °C.

## 4. Discussion

VBNC status is a survival mechanism commonly adopted by microorganisms in response to external stress, and more than 60 species of VBNC bacteria have been reported to employ this strategy, including various human pathogens [40,41]. Studies have highlighted the close link between the efficacy of various disinfection methods and the intrinsic characteristics of the target bacteria, as well as the composition and concentration of disinfectants used in Table S5. In the single disinfection method, $UV_{254}$ had the highest inactivation rate of live and culturable bacteria (K = 0.059), followed by NaClO and PAA treatment. UV/NaClO combined disinfection method (K = 0.2642) had obvious advantages in removing culturable bacteria and reducing the VBNC state. Similar effects have been shown in other combined processes such as UV/PDS and UV/PAA, and this benefit can be attributed to the production of potent free radicals (e.g., OH•, $SO_4$•$^{2-}$, Cl•, NaClO•$^-$, and ClO•) in the UV-AOPs. Overall, enhanced UV intensity and optimal initial oxidizer concentration were positively correlated with disinfection effectiveness [42]. In addition, water quality parameters such as turbidity, pH and organic matter content have a great influence on the disinfection effect of UV-AOPs. pH has an effect on chlorine dissociation (HOCl/OCl [pKa = 7.5]) [43] and pathogen activity. The presence of organic matter significantly hinders the efficacy of UV-AOPs by absorbing ultraviolet radiation, thereby reducing the number of photons entering the reaction solution and reducing the direct photolysis rate of pathogenic microorganisms [44]. The influence of other water quality parameters (temperature, water age/stagnation, disinfectant residue, or presence of soil and sediment) and building water system design characteristics on the disinfection efficiency of the UV-AOPs combination needs to be further explored.

The aim of this study was to assess the viability of UV/NaClO, UV/PDS, and UV/PAA combined inactivation strategies against *P. aeruginosa*. While this research study introduced some innovative aspects, it also presents several limitations warranting further investigation. The results of this study from a single laboratory experiment (bench science) are not yet translatable to practical field applications for reducing VBNC bacteria reactivation, especially in the context of civil engineering or building engineering applications. Additional research will require a systematic approach to determine validity and practicality. Basic steps in this process include field validation, engineering suitability assessment, compliance with regulatory and standard protocols, risk assessment and mitigation, and provision of training and education. Collaboration with relevant stakeholders, including government agencies, industry associations, and businesses, is essential to ensure the effective translation and use of research findings. The application of combined disinfection methods occurs

mainly in the final stages of water treatment plants. This ensures a lower risk and reduced exposure to VBNC *P. aeruginosa*, thereby mitigating potential public health hazards. In specific, we see future research in the following aspects:

1.  Methodological refinement is necessary. Although utilizing longer gene segments, our study more accurately quantified the cell count of VBNC P. aeruginosa after disinfection. The PMA-qPCR method still presents other challenges in assessing the VBNC pathogens in water. Firstly, the PMA-qPCR process involves multiple steps, including sample pretreatment, DNA extraction, PMA treatment, and PCR amplification, necessitating automation for efficiency. Secondly, the sensitivity of the PMA-qPCR method, currently limited to 3.191 log CFU/mL, varies depending on factors such as target DNA concentration, PMA penetration efficiency, and sample inhibitors. Hence, there is a need to optimize detection methods for enhanced accuracy and convenience.

2.  Enhancing the depth of research is imperative. The probe method and quenching method were used to explore the disinfection mechanism of UV-AOPs combined disinfection, which needs to be further explored. Future research should integrate transcriptomic, proteomic, and metabolomic analyses to elucidate bacterial response mechanisms to disinfectants, providing insight into the inactivation mechanism of *P. aeruginosa* by UV-AOPs combined disinfection.

3.  Expanding research breadth is essential. While this study systematically examines the disinfection effects and mechanisms of three UV-AOPs combined disinfection treatments on VBNC *P. aeruginosa*, broadening the research scope is crucial. Future studies should encompass additional pathogenic bacteria such as *Legionella*, include environmental factors like water quality parameters (pH, temperature, water age/stasis, disinfectant residue, or soil and sediment presence), and explore diverse scenarios, such as city utility water processing or in high-risk building water distribution systems.

## 5. Conclusions

Bacterial reactivation after disinfection in drinking water systems has always been a pressing public concern. This phenomenon is attributed to the disinfection-induced VBNC state, where microorganisms are not directly eliminated but rather experience reduced biological activities that hinder their reproduction. This study introduces a combined disinfectant strategy by integrating $UV_{254}$ with disinfectant for effectively inactivating *P. aeruginosa*. This study found that the UV/NaClO combination disinfection method for 30 min of treatment reduced the presence of *P. aeruginosa* without reforming a VBNC state after 16 h of time under controlled laboratory conditions. Other combinations, including UV/PDS and UV/PAA, also significantly reduced the number of VBNC cells by several orders of magnitude. Probe and quenching experiments revealed that suppressing the reactivation of VBNC *P. aeruginosa* by UV/disinfectant exposure for 30 min was ascribed to the generation of reactive radicals in AOPs. Although the turbidity in water matrices can decrease the disinfectant performance for *P. aeruginosa*, the extension of the exposure time can overcome this negative effect. These findings significantly advance our understanding of the disinfection mechanisms of *P. aeruginosa* by UV-AOPs and provide important insights into its future application in drinking water purification technology.

**Supplementary Materials:** The following supporting information can be downloaded at https://www.mdpi.com/article/10.3390/w16091302/s1. Table S1. Primers for the amplification of the opr gene; Table S2. Second-order rate constants k ($M^{-1}$ $s^{-1}$) for probe compounds and free radical species; Table S3. UPLC operation conditions for NB; Table S4. UPLC operation conditions for MET; Table S5. The disinfection effect of different disinfection methods on different pathogenic bacteria; Figure S1. Water turbidity in spring and summer at each sampling point in Shanghai, China; Figure S2. Inactivation rate constants of P. aeruginosa under single disinfection method; Figure S3. Inactivation rate constants of P. aeruginosa under combined disinfection; Figure S4. Inactivation curves

after the addition of excess TBA; Figure S5. Degradation of NB (a) and MET (b) by UV/NaClO, UV/PDS, UV/PAA, and UV.

**Author Contributions:** Conceptualization, J.Z., H.Z., C.T. and Z.W.; methodology, J.Z., H.Z., C.T., N.D. and X.H.; investigation, J.Z., H.Z. and C.T.; resources, N.D., Z.W. and X.H.; data curation, J.Z., H.Z. and C.T.; writing—original draft preparation, J.Z., H.Z. and C.T.; writing—review and editing, H.Z., N.D. and X.H.; supervision, X.H. All authors have read and agreed to the published version of the manuscript.

**Funding:** This research was funded by the National Natural Science Foundation of China, grant number No. 51678351.

**Data Availability Statement:** All data were published in the main text and Supplementary Materials.

**Conflicts of Interest:** The authors declare no conflicts of interest.

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
