# Peer review of "A Combination of UV and Disinfectant for Inactivating Viable but Nonculturable State Pseudomonas aeruginosa: Efficiency and Mechanisms"

_water, doi:10.3390/w16091302_

Round 1

Reviewer 1 Report

Comments and Suggestions for Authors

The paper presented by Zhao et al. entitled "Combination of UV And Disinfectant for Inactivating VBNC State Pseudomonas aeruginosa: Efficiency and Mechanism" is relevant to the scope of the journal. The manuscript provides important insights and data on the mechanism of advanced oxidation processes (UV-AOPs) for the inactivation of VBNC state bacteria in water. However, there is still room for improvement in terms of organization, language expression, and data interpretation. major revision is needed before the consideration for publication

1. The names of bacteria in the paper should be used in italics, such as in lines 32 and 36. 

2. In lines 87, the abbreviations PAA and PDS were used without their full forms, which reduced the readability of the article.

3. Why not use the same molar concentration as PAA and PDS for NaClO? The different initial concentrations of the three disinfectants may affect the rate of generation and concentration of free radical, resulting in different inactivation effects.

4. The effect of factors such as the quantum yield and free radical generation efficiency of NaClO, PDS and PAA on the inactivation effect should also be considered.

5. Statistical analyses are missing from the descriptions of data results in the articles.

6. The full form of the VBNC for the first time in the article, and the abbreviation will be used consistently thereafter, such as lines 156, 217 and 227.

7. In the article, UV radiation dose selected is much higher than the actual radiation dose used in water treatment scenariosThis is not conducive to provide important insights for drinking water treatment.

8. In lines 221, "This indicated that P. aeruginosa was directly eliminated by UV/NaClO treatment without a VBNC state." The detection limit of PMA-qPCR for VBNC is 3.191 log CFU/mL. Therefore, the results can only indicate that the number of VBNC bacteria after UV/NaClO treatment is lower than that of UV/PDS and UV/PAA, but it cannot fully confirm P. aeruginosa without a VBNC state after UV/NaClO treatment.

9. The effect of turbidity on inactivation is discussed in section 3.4 and the reagents to be used to configure the analogue solution (40 NTU) should be described in the experimental methodology.

10. Turbidity can affect the transmittance of UV radiation, so there may be significant differences in the actual dose of UV radiation received for the same irradiation time. Different irradiation doses have different effects on bacterial inactivation. Therefore, a comparison of the inactivation effect at the same irradiation dose would be more convincing.

11. As is well known, the integrity of cell structure and metabolic activity are closely related to VBNC state bacteria. The author should provide a more detailed comparison from the more perspectives, such as membrane integrity, ATP, respiration activityetc.

12. For Fig. 3, "Contribution (%)" would be more appropriate for the vertical coordinates.

13.  The colors and formatting of all the figures need to be improved.

Comments on the Quality of English Language

 Moderate editing of English language required

Reviewer 2 Report

Comments and Suggestions for Authors

Some remarks are given to improve this manuscript.

Throughout the manuscript, the term “resuscitate” is used. Since microorganisms are inactive state (not death), I consider that this change should be changed.

Lines 49 – 59. Several references are needed.

Lines 64 – 66. Reference [20] is similar to the aim of the manuscript. Please, describe the differences between the metabolic activity and dark reactivation analysis carried out by [20] and the present study. I understand that this manuscript employed the PMA-qPCD analysis to assess the viable and culturable cells in VBNC state of P. aeruginosa, however, similar results are obtained. Please, highlight the differences and novelty of the work.

Lines 82-84. Can this innovative method be comparable with "traditional methods"?

Can this innovative method be continuous applied for monitoring the drinking water facilities? Please, discuss technical and economic advantages and disadvantages of the proposed methodology to assess the VBNC state of the pathogens in water.

According to equation 3, inactivation kinetics for all methods (UV, NaClO, PDS, PAA, among others) follow a pseudo first order model. A justification of this must be included.

According to Figure 1, the methods were carried out separately. However, according to equation 3, the implementation of all methods in one experiment is required, starting one after having finished a previous one. Why were these methods tested separately? This situation must be explained.

Was an experimental design used to assess the synergistic effect? An in-depth explanation of the experiments must be provided to better understand the results presented in section 3.

According to Figure 1, the UV/NaClO treatment shows values below the detection limit of the technique. Since other methods must be used after UV/NaClO treatment to meet equation 3, how can viable cells be measured?

Figure 2. Can the authors provide a figure to understand how these kinetics rates were obtained? Please, include the coefficient of determination (r2) for each graph.

Lines 245- 247. These rate values must be compared with other studies in the literature.  

Lines 273 – 279. How were these contributions (percentages) obtained? Please, include a better description of this.

Reactivation test after UV/ NaClO, UV/PDS, and UV/PAA treatments were not explained in section 2.

Reviewer 3 Report

Comments and Suggestions for Authors

Overall, the peer-reviewer finds the study very interesting and potentially novel in approach. However, the write up of the study is missing key elements of presenting scientific evidence such as incomplete introduction/background, lack of citing prior methods used, and lacks a separate formal discussion section of findings stated in the context of other research.  Also, the authors do not explain how this small scale laboratory experiment would be translated into any sort of real world application for water disinfection.  Water is not disinfected in a  laboratory "bench" like setting.  Water disinfection and hazard control treatments are applied through highly complex civil engineering and/or building engineering water distribution systems.  Something that works in a laboratory may or may not work under real world application. The study does not have a limitations section and the study does not suggest next steps to proceed to a either increase the scale of the study (replication or advancing study methods) or describing a real world level of application. 

Additionally Limited use of citations:  There are 24 citations.  A high percentage  (21 of 24 - 87%) of the citations are for the scientific introduction/background with only a couple in the materials and methods, and a couple in the results.  There are many missing citations. By developing a formal discussion section there should be appropriate number and different citations to contextualize the authors' research and findings to current science on this topic.

Specific comments include:

Abstract:  After all the other items are addressed in the peer-review, the authors need to return to the abstract and update this section according to the new writings for discussion, limitations, and conclusions.

Introduction:

1) Epidemiology statement.  How large is this problem? in the first paragraph there is no epidemiological information to suggest size of the problem. The readers need to know how many annual P. aeruginosa disease cases (China, Asian countries, USA or other world sectors for this pathogen of interest) exist and where (healthcare) and what are the likely routes of transmission (contaminated hands, medical equipment, surface transmission)?  The reader has no idea from the description how common or high risk this situation is.  Please provide one or two sentences to for epi- context after line 38/39.

2) Introduction Line 45 to 46: respectfully suggest the authors update language here removing casual wording "this however is not the end of the story" and restructure the sentence to be more scientifically stated such as:  However, this is not the end of the pathogen's lifecycle since VBNC P. aeruginosa can reactivate when environmental........

3) Introduction: The aims paragraph needs to be expanded and clarified.  here are some suggestions to do that:  After the study's aims are stated on lines 86 - 89, the authors should insert a sentence about the novelty of this study vs replication of past studies.  The reader cannot clearly discern where the authors are in the line of the "arc of research" for waterborne pathogens, VBNC, and UV light over time.  Why is this study unique vs. other studies on UV light treatment or other studies on combining water treatments?  That is not clearly expressed in the document but more assumed this is somehow novel. It would benefit the authors to have an expressed statement of research novelty.  The authors should also state a clear research question(s) they are trying to answer or hypothesis statement?

4) Introduction: It seems unusual that Legionella is not expressly mentioned in the introduction as a similar situation to P. aeruginosa as a waterborne pathogen that also is struggling with VBNC status, and has also had studies on UV light and many other disinfection similarities.  Reference 8 does mention Legionella in its title, but it seems like a missed opportunity to mention VBNC in the context of similar pathogens of interest with exactly the same challenge.  Authors should mention similarities to Legionella in the introductory paragraph lines 31 - 38.  This will likely be important related to other comments the authors need to resolve in the peer-review pertaining to a) writing a new Discussion section, b) writing a new limitations section,  and c) improving the conclusion section.

5) UV Light vs UV irradiation:  This might be my lack of knowledge on UV terminology but as a reader I found the use of different terms for what I believe is the same thing confusing throughout the article.  Is the disinfection treatment "UV light" at a defined wavelength (254 nm)? Or is it UV irradiation? are these the same process/terms? The authors should better explain UV light vs. UV irradiation or the UV process as a whole in the paragraph line 49 - to 59 to set the reader up for ONE definition or term throughout the article. 

6) Introduction lines 73 - 84: there are no citations for these statements verifying and validating these scientific methods stated.  If the authors recently developed a technique, they should have published that method and cite it prior to submitting this publication; or if they based their process on someone else's process then that needs to be cited.  If these prior study methods have never been published, the authors should likely not present this research on unpublished methods? Please provide citations and/or clarifications. 

Materials and Methods Section:

1) Laboratory Setting:  insert some description of the analytical laboratory and its location, equipment, or certifications to conduct this sort of experiment.  Suggest renumbering and having 2.1 Laboratory Setting and tell the reader what makes this laboratory environment, location, and staff capable of responsibly handling these types of pathogens and disinfection methodologies.  Such as is this a commercial analytical laboratory or academic university laboratory?  What health or safety laboratory certifications has the laboratory institution obtained? what training does the staff have to assure these pathogens and the equipment is authorized, handled properly, equipment is calibrated annually? This doesn't have to be a long text but the reader should be given a statement as to who handled these samples and what environmental controls are in place and necessary to conduct such an experiment.  In most countries analytical laboratories would have to be authorized to conduct such experiments with specific certification, accreditation, or other independently verified laboratory conditions and staff training in place to assure responsible procedures are followed.  Please provide such statements appropriate for this country and setting.  

2) Line 120:  Previous studies [INSERT CITATION] have reported that UV irradiation....    Authors need to carefully review the entire methods section and insert citations for the methods and processes undertaken.  This is one example of a lack of proper citations. 

3) Clearly articulate the number of samples that were tested?  From the methods description under Subsection 2.5, it sounds like the authors tested one sample? and then the text refers to each group of samples ( how many) was subjected to ten parallel sets and all counts were executed in triplicate.  My question is how many samples were actually exposed to the UV irradiation + combination treatment?  Is this entire study based on one sample per disinfection method? Or where is that clearly explained.  The authors should consider a diagram that explains the sampling process and UV irradiation + disinfection process. 

Results -

1) Results need to be separated from the Discussion (see Discussion section below).  I suggest leaving the current results section as is and write a new Discussion section.  Since the authors are not presenting traditional statistical analysis to demonstrate significance (0.05) of the intervention, the authors need some clear explanation in a discussion section why are these results significant or not significant.

2) All the figure(s) labels and descriptions need to be updated to MDPI formatting.  The location of Figure 1, 2, 4, and 5 including a) description and b) description and c) description are not organized correctly.  The figures and titles are spread over multiple pages for one figure grouping making reading the figure analysis very difficult for the reader. 

3) Time on X axis for Figures- time on some figures is in seconds and others hours.  Figure 4 shifts to hours - is this correct? yet the sample title is 30 minutes? This seems confusing.  The reader understands from the text writing that reactivation occurred after hours of time.  I think the title of these should consistently read: Reactivation (time in hours) of P. aeruginosa after 30 min of treatment with a) UV/NaCIO, b) UV/PDS, c) UV/PAA.  In the figures themselves I would also include after 30-min treatment.  When the reader first glances at these figures, what they are seeing is confusing.

Discussion

The peer-reviewer recommends the authors need to read more literature to determine what studies they want to contextualize their research within.  Below are a series of suggested questions the authors need to consider.   Overall the authors need to explain 1) why is this type of "bench science" that diminishes the likelihood of VBNC resuscitation significant?  and 2) how do we advance and translate this small scale laboratory experiment forward toward a real world application.  See below......

1) The authors provide no real discussion of comparing the results of this study to other studies and findings to set this research in context.  The authors have an obligation to the scientific process to contextualize their findings with waterborne pathogens, UV treatments, and combining treatments.  As mentioned earlier, other pathogens like Legionella are also studying the impacts of VBNC.  How does this study compare to other findings for Legionella VBNC? All the the articles citations are primarily in the background section and a few in the methods section.  New literature needs to be read and cited.

2) Further the authors need to clearly differentiate their findings conducted in an analytical laboratory setting under controlled conditions to water samples versus water that is circulating in large civil engineering systems (city water) and building water distribution systems (hospitals, schools, high rise residences, etc). The authors mention turbidity as one aspect of water quality that can adversely influence disinfection performance. What about the many other water quality parameters (pH, temperature - hot vs cold water, water age/stagnation, disinfectant residual, or the presence of soil and sediment)? or building water system design features (different piping materials, piping size, velocity of water distribution, water efficiency or conservation, etc.)? How do these common water parameters impact UV/NaCIO, UV/PDS, or UV/PAA disinfection methods tested.  If these parameters were not studied here, then these are limitations of the study and the findings need to be stated as such.  All of these factors that are mentioned act like confounding variables that were potentially not researched and therefore the findings need to state that. 

3) How does research like this study move from analytical laboratory setting (bench science) to real world (field - civil engineering or building engineering) application?  What regulations or testing procedures would need to be in place to assure that this type of hazard control worked in a field setting?  Where in the water distribution system would this combined disinfection process take place? or  be located? at the water main point of entry, on cold water distribution supply, or after hot water heating return loop system, or other?  

Limitations

1) Once the authors establish the topics of the discussion section, they need to write a limitations section.  What does this study cover and what does it not cover?  Researchers need to guard against misinterpretation of the findings for the wrong purpose or for an inappropriate application by stating the limitations of the study here?  Examples from above are if you have no idea how UV/disinfection combinations will work in real world field applications or under hot water conditions or any of the other commonly studies water parameters and piping distribution settings, then the authors need to state the limitations of the study and its intended application.  Research studies can't cover everything and cannot be generalizable to all applications especially with new bench science research.  The article reads as if this is a novel research method and application at the bench science level without restrictions or thoughts regarding real world application without further research being conducted. 

Conclusion.  These conclusions are very broad and need to be narrowed. 

1) Sentence line 333 - 335 needs narrowed to the findings of this study and not stated as generalizable to all environments and conditions since the authors didn't present evidence supporting wide spread application.  Example:  This study found that UV/NaCIO combination disinfection method directly eliminated P. aeruginosa without forming a VBNC state after 16 hours of time under controlled laboratory conditions. 

2) somewhere in the conclusion - further studies are necessary to replicate these findings to assure the VBNC status does not resuscitate over longer time periods such as days (24 hours) or weeks (7 days), or X months, etc. More studies using this approach are necessary to replicate findings and determine longer time periods for VBNC resuscitation?

Then depending upon the authors Discussion section they might need to formulate sentences about future research.  Some examples might be:

3) further studies are also necessary to determine if this works under various field conditions for city utility water processing or in high risk building water distribution systems such as hospitals. 

4) should/can anything from this research be applicable to Legionella spp. for trying to control VBNC conditions for other troubling waterborne pathogen growth and spread in city water or building water systems?

Comments on the Quality of English Language

Series of editorial text and organizational comments to make the article read easily.

1) Pseudomonas aeruginosa and P. aeruginosa - please search the entire article and make sure all pathogens are italicized properly.  Some are and some are not.  I found pathogens that were not italics on lines 36, 37, 41, and 229.

2) Chemical, Pathogen, Analytical Testing Acronyms - the article by necessity  has a large number of scientific acronyms.  That's fine but it is very confusing based on the current writing.  Many acronyms are not explained at their first mention making the reader trying to search for meanings and figure this out on their own. The abstract should not be used for first defining these versus in the text upon first appearance.  Some example are:

- HPC - first mentioned on line 81 but not defined until line 131; please change that and then go through the article and fix and repair each of these situations as an example on line 131, 197.

- similar situation with NaCIO (sodium hypochlorite) first mentioned on line 64 but not defined until line 98. 

To fix this acronym situation overall, the authors need to work with the publishing editor and resolve this confusion.  As a reader I was flipping through pages of trying to figure out which definition belonged with which disinfection method.  My respectful suggestion might be to improve the introduction of describing existing disinfection methods early on and refer the reader to the reagents section of the article for all the additional definitions the authors are mentioning.  Example:

Line 49: Understanding the limitations of existing disinfection methods (sodium hypochlorite - NaCIO, peracetic acid - PCC, and potassium persulfate - PDS) can realize an improvement in inactivating bacteria (for other reagent or chemical definitions see Section 2.1 Reagents). 

3) Line 50- 51: Conventional drinking water treatment commonly employs a single disinfection method, such as application of ultraviolet (UV) and or chlorination.  [the authors suggest single disinfection method and then follow the sentence by suggesting two disinfection methods together.  I think you mean the use of the word  "or" not "and" in this context.] 

4) Check comments from author section comments on fixing figures to read in MDPI format with respect to naming conventions for multiple figures (a, b, c) within one Figure section.  The titles and descriptions were not labeled well and very difficult to navigate in the current manuscript submission.

Round 2

Reviewer 1 Report

Comments and Suggestions for Authors

The author has carefully revised the article, but there are still some issues that require further corrections.

1.      Please check and remove extra punctuation, e.g. line 149, 317, etc.

2.      In line 185, "VBNC - state P. aeruginosa", it is recommended that the "-" be removed.

3.      Line 211, P. Aeroginosa should be rewritten as P. aeruginosa.

4.      In line 248-249, “the inactivation of VBNC bacteria was greatly enhanced by UV/NaClO treatment”. Suggest modifying this sentence. For example, “UV/NaClO treatment can effectively control P. aeruginosa to enter VBNC state”.

5.      In line 248-249, “This suggests that combined UV/NaClO disinfection directly inactivates VBNC P. aeruginosa without VBNC cells formation.” It is proposed to delete the first VBNC in this sentence.

6.      As mentioned in the paper, differences in water quality can affect the dose of UV radiation. Therefore, for different samples with significant differences in water quality, the same UV irradiation time will ultimately result in different irradiation doses. At present, the national standards for UV disinfection are based on the dosage of ultraviolet radiation, and the content related to UV disinfection experiments in the paper is not appropriate to use the unit of irradiation time.

7.      The data in Table 1 of the article should be clearly identified as to which are experimental results and which are quotes from others.

8.      Suggestion for removal of redundant connection lines in Figure 3.

9.      The references should be carefully checked, such as the italicization of bacterial names and the subscript of NH2Cl, etc.

Comments on the Quality of English Language

Minor editing of English language required

Reviewer 2 Report

Comments and Suggestions for Authors

I would like to congratulate the authors of the manuscript for the excellent work done during the review stage. The answers provided address all the remarks. No further comments are given.

Author Response

We greatly appreciate your time and consideration throughout the review process. We will ensure to follow the instructions provided by the journal editor for the next steps in the publication process.

Thank you once again for your invaluable support.

Reviewer 3 Report

Comments and Suggestions for Authors

Overview and general comments

The authors have responded to the comments and the answers were appreciated.  The corrections provided (from the prior first round of comments) are accepted by the reviewer with the following exceptions noted below that are a combination of new comments and clarifications on comments that were not entirely addressed. 

In general, there are two areas of continued concern.  The authors keep expressing results using adjectives or absolutes such as “surprisingly” or “completely eliminate” or “eliminated almost all” and or that VBNC P. aeruginosa was eliminated.  Water and these pathogens are organic material that can change over time.  The results have to be expressed in written form using data not adjectives or absolutes. 

Additionally, the authors seem to believe that writing extensive results is also the discussion section.  The Discussion section in scientific publications is where the authors compare and contrast their findings to other studies published on similar topics in this case disinfection and/or dual disinfection.  For additional information about this see Specific Comment Line 219.  The final decision will have to be made by the MDPI Water editorial staff.    

 Specific Comments

Lines 21 – 26:  Abstract: The authors have over generalized their findings using phrasing like “completely eliminated” or “no P. Aeruginosa”.  It was suggested prior that all statements for findings need context and should be stated using narrower language (actual data) to express findings.

Quantitative analyses showed that combined disinfection methods can effectively reduce both culturable and VBNC cells by several orders of magnitude compared to single disinfection step. Notably, VBNC P. aeruginosa after 30 minutes of UV/NaCIO treatment was below the detection limit (3.191 log CFU/mL) of PMA-qPCR.  The reactivation experiment also confirmed that VBNC P. aeruginosa did not reactivate for 16 hours after 30 minutes of UV/NaClO treatment under controlled laboratory conditions. The higher disinfection capacity of combined methods can be attributed to the generation of reactive radicals. This study highlighted the combined disinfection as a promising approach for the inactivation of bacteria in the VBNC state, yet further studies are needed before application is considered for minimizing VBNC reactivation in city utility water processing or high-risk building water distribution systems.   

Line 27-28:  Add to keywords to make this study and its results more searchable:      disinfection, VBNC; water safety 

Lines 40 – 41: check sentence and acronyms for capitalization (e.g., settings and ICUs)

Lines 53 – 64:  insert your explanatory text to the peer-reviewer to explain UV light and its application to the reader.  Please modify/edit to incorporate around the surrounding sentences. 

Ultraviolet (UV) light refers to the emission of light from UV lamps, while UV irradiation denotes the exposure of a substance to UV light. It is important to note that UV light encompasses a range of wavelengths, including 222 nm, 254 nm, 275 nm and 405 nm. The deep ultraviolet band of 254 nm UV (UV254) has the best bactericidal effect, and is widely used in real application. To facilitate our study UV25 will uniformly represent the UV treatment method.

Lines 81 -83:  insert a citation of the study mentioned.  It appears it is [currently #22?] but your team will need to check citations numbering if necessary.  

Lines 106 to 113:  Since a specific certification for the laboratory was not listed in Section 2.1 some statement needs to be added related to laboratory equipment used in the study.  Example: 

All laboratory equipment (e.g., centrifuges, UV254 lamp) for the study was tested and calibrated to be operating per manufacturers’ requirements prior to conducting the study for collecting data. 

Line 141:  Section 2.4 – authors state “Previous studies [INSERT CITATIONS] have reported that UV254…..”

What studies are the author’s referring to?  

Line 219: Results and Discussion – the authors have stated that the language in the Results section is also the Discussion section.  Respectfully, this is confusing to the peer-reviewer.  Writing extensive text about the study’s findings to explain them to the reader is not the purpose of the Discussion section.  A discussion section in scientific literature is a separate section in which the authors discuss their findings in the context of prior published studies on this subject or related subjects.  It is to give context to the authors’ findings in order to compare and contrast the study’s findings to those of other studies who might have combined disinfection methods and obtained similar or different results.  Are the author’s findings normal or abnormal to previous scientific studies?

Based on the current manuscript, the authors leave it up to the reader to have to go search for additional studies and figure out if their findings are similar or different from others who have combined UV light with a second disinfection method and is this already done in the industry in general but not yet for VBNC purposes? From my point of view this approach is an incomplete analysis of the science. I don’t know if MDPI Water publication wants to accept the author’s point of view to essentially have an incomplete discussion section or not?  I will leave that up to the Water editorial team to decide.   

Line 233, 275, 338, 360 MDPI Figure labels:  These figures are still incorrect.  I will let the MDPI Water editor deal with the authors on getting all these figures fixed.  But simply note that all lettering of the figures (a, b, c, etc) are to be underneath each figure --- not to the left, to the right, or inside the figure itself.  All figure labeling is currently inconsistent and confusing to the reader to find; figure labels should use the same method. 

Line 322 – 325:  This sentence is awkwardly worded and uses inappropriate phrases such as:

“Surprisingly”, UV/NaCIO “eliminated almost all” P. aeruginosa and “completely suppressed” its reactivation within 16 hours.  Results should not have adjectives and other terminology that leads the reader to conclusions that are not data driven.  Statements like these are inappropriate and inaccurate.  Need to change this to read factually based on data. Example phrasing:

P. aeruginosa after 30 minutes of exposure to UV/NaCIO remained suppressed [INERT number found 0.0 Log10 (cells/mL) for up to 16 hours under controlled laboratory conditions (Figure 4a).  This level of detection suggests that combined UV/NaCIO disinfection inactivated VBNC P. aeruginosa without VBNC cells formation for up to 16 hours. 

Line 371 – this line was listed in the prior 1st round review comments and the authors in their response agreed to change this language.  However, the text was not updated.  Suggested new phrasing:

This study found that UV/NaClO combination disinfection method for 30 minutes of treatment reduced the presence of P. aeruginosa without reforming a VBNC state after 16 hours of time under controlled laboratory conditions. 

Line 374 – the use of the word “eliminating” is not appropriate.  This word suggests that the pathogen will never reappear which the study did not find and cannot assure.  Other suggested language might be:

Probe and quenching experiments revealed that suppressing reactivation of VBNC P. aeruginosa by UV/disinfectant exposure for 30 minutes was ascribed to the generation of reactive radicals in AOPs.

Line 381 Limitations: Limitations are not part of Conclusions.  They are typically stated at the end of the Discussion section to close out what was not covered in this study.  If the Discussion section MDPI Water editors decision) remains part of the Results Section then it would appear as Section 3.5 Limitations. 

-        Text needs to be moved forward. 

-        The authors need an introductory first paragraph and then can use the text that they have already developed in the response to the reviewer.  I provided some limited wordsmithing to assist the authors and assist in striking language that is inaccurate to claim (e.g., complete elimination, etc.)

-        Suggest using the text written in their response to the peer-review comments as the first paragraph

The aim of this study was to assess the viability of UV/NaClO, UV/PDS, and UV/PAA combined inactivation strategies against P. aeruginosa. While this research study introduced some innovative aspects, it also presents several limitations warranting further investigation.  The authors are aware that our study’s results from a single laboratory experiment (bench science) are not yet translatable to practical field applications for reducing VBNC reactivation, especially in the context of civil engineering or building engineering applications.  Additional research will require a systematic approach to determine validity and practicality. Basic steps in this process include field validation, engineering suitability assessment, compliance with regulatory and standard protocols, risk assessment and mitigation, and provision of training and education. Collaboration with relevant stakeholders, including government agencies, industry associations and businesses, is essential to ensure the effective translation and use of research findings. The application of combined disinfection methods occurs mainly in the final stages of water treatment plants, ensuring lowering risk and the reduced exposure to VBNC P. aeruginosa, thereby mitigating potential public health hazards. In specific we see future research in:

1.      Methodological refinement…….

2.      Enhancing the depth of research……….

3.      Expanding research breadth……….

Comments on the Quality of English Language

The English needs minor editing.  There are minor spacing, capitalization issues, and other grammar items.  Reviewer made some comments but most of this can be dealt with through the journal's editorial final review process.  Please continue to fix the figures that are not labeled in MDPI format. 
